# WNK regulates Wnt signalling and β-Catenin levels by interfering with the interaction between β-Catenin and GID

Atsushi Sato [1,4,5], Masahiro Shimizu [1,2,5], Toshiyasu Goto[1], Hiroyuki Masuno[3], Hiroyuki Kagechika[3], Nobuyuki Tanaka[2] & Hiroshi Shibuya [1✉]

β-Catenin is an important component of the Wnt signalling pathway. As dysregulation or mutation of this pathway causes many diseases, including cancer, the β-Catenin level is carefully regulated by the destruction complex in the Wnt signalling pathway. However, the mechanisms underlying the regulation of β-Catenin ubiquitination and degradation remain unclear. Here, we find that WNK (With No Lysine [K]) kinase is a potential regulator of the Wnt signalling pathway. We show that WNK protects the interaction between β-Catenin and the Glucose-Induced degradation Deficient (GID) complex, which includes an E3 ubiquitin ligase targeting β-Catenin, and that WNK regulates the β-Catenin level. Furthermore, we show that WNK inhibitors induced β-Catenin degradation and that one of these inhibitors suppressed xenograft tumour development in mice. These results suggest that WNK is a previously unrecognized regulator of β-Catenin and a therapeutic target of cancer.

[1] Department of Molecular Cell Biology, Medical Research Institute (MRI), Tokyo Medical and Dental University (TMDU), Bunkyo-ku, Tokyo 113-8510, Japan. [2] Department of Molecular Oncology, Institute for Advanced Medical Sciences, Nippon Medical School, Bunkyo-ku, Tokyo 113-8602, Japan. [3] Institute of Biomaterials and Bioengineering, TMDU, Bunkyo-ku, Tokyo 113-8510, Japan. [4] Present address: Department of Biological Sciences, Graduate School of Science, The University of Tokyo, Bunkyo-ku, Tokyo 113-0032, Japan. [5] These authors contributed equally: Atsushi Sato, Masahiro Shimizu. ✉email: shibuya.mcb@mri.tmd.ac.jp

The Wnt signalling pathway is an essential pathway in multicellular organisms that has already been identified to function in cell proliferation, differentiation and home-ostasis[1–3]. Dysregulation or mutations of this pathway are related to many diseases, such as type II diabetes[4], autosomal dominant polycystic kidney[5] and many cancer types[6]. The Wnt signalling pathway is divided into three classes: the canonical pathway, planar cell polarity pathway and Ca pathway. In the canonical Wnt pathway, β-Catenin is a core component whose protein regulation is a key process in the Wnt signalling pathway. In the absence of Wnt, the destruction complex formed by Axin1, Adenomatous Polyposis Coli (APC) and Glycogen Synthase Kinase 3β (GSK3β) phosphorylates the N-terminus of β-Catenin, which is in turn recognized and ubiquitinated by βTrCP E3 ubiquitin ligases[6,7]. Ubiquitinated β-Catenin is degraded by the proteasome, which regulates the basal β-Catenin levels. The Wnt signalling pathway is activated by the binding of Wnt ligand to a receptor complex[8,9], which leads to Dishevelled (DVL) recruit-ment, where DVL polymerizes with Axin and the remaining components of the destruction complex. This polymerization inactivates the destruction complex, inducing β-Catenin accu-mulation in the cytoplasm. Accumulated β-Catenin translocates to the nucleus and regulates the transcription of target genes[10]. However, because several ubiquitination enzymes of β-Catenin have been identified and shown to be Wnt-independent[11–17], the mechanisms underlying the regulation of β-Catenin ubiquitina-tion remain unclear.

The WNK (With No Lysine [K]) kinases are atypical serine/threonine kinase family members that are conserved among many species[18–20] and comprise four members. WNK1 and WNK4 have been identified as the causative genes of pseudohy-poaldosteronism type II[21], and WNK1 is also a causative gene of hereditary sensory and autonomic neuropathy type 2A[22]. WNK kinases are required for epidermal growth factor-mediated extracellular signal-regulated kinase 5 activation, and WNK family members are also involved in proliferation, migration and differentiation[23–25]. Previously, we found that WNK1 and WNK4 induce the expression of Lhx8 and are important for neural specification through GSK3β[26,27]. WNK was also identified as a positive regulator of the Wnt signalling pathway; however, the detailed mechanism is unknown[28]. Thus, WNK has several functions associated with many developmental processes.

Here, we demonstrated that WNK is a positive regulator of the Wnt signalling pathway. WNK attenuates the interaction between β-Catenin and the glucose-induced degradation deficient (GID) complex, which is an E3 ubiquitin ligase of β-Catenin, suggesting that WNK might regulate β-Catenin levels. Furthermore, we showed that the WNK inhibitor also functions as a Wnt inhibitor, suppressing xenograft tumour development in mice. These find-ings suggest that WNK is a regulator of β-Catenin and might be a therapeutic cancer target.

## Results

### WNK functions as a positive regulator of the Wnt signalling pathway.

We previously showed that GSK3β, a component of the Wnt signalling pathway, functions downstream of the WNK signalling pathway[27]. GSK3β acts as a negative regulator of the Wnt signalling pathway[6,7]. Additionally, WNK was shown to be involved in the phosphorylation of Dsh (fly homologue of DVL) under Wnt stimulation in Drosophila and to work as a positive regulator of the Wnt signalling pathway in HEK293T cells[28]. Thus, we considered whether WNK interacts with the Wnt sig-nalling pathway. First, we examined whether the knockdown of WNK affects the Wnt signalling pathway. We performed RT-PCR to examine the expression of the Wnt target genes AXIN2

(early response gene) and c-Jun (late response gene) under the knockdown of WNK1 and WNK4 in HEK293T cells. We found that the expression levels of AXIN2 and c-Jun were activated by Wnt stimulation (Fig. 1a). The knockdown of both WNK1 and WNK4 using siRNA significantly reduced the induction of both AXIN2 and c-Jun (Fig. 1a). These results suggest that WNK is a positive regulator in the Wnt signalling pathway.

WNK is thought to be a cytoplasmic protein[20], and therefore we next analysed the relationship between WNK and DVL, the most upstream cytoplasmic component of the Wnt signalling pathway. We found that DVL1 expression activates the Wnt signalling pathway[6,7] (Fig. 1b). The knockdown of both WNK1 and WNK4 suppressed the expression of AXIN2 and c-Jun activated by DVL1 expression (Fig. 1b). Exogenous expression of all DVLs (DVL1, DVL2 and DVL3) did not rescue the suppression of Wnt3a-activated gene expression of AXIN2 and c-Jun by the knockdown of both WNK1 and WNK4 (Fig. 1c). Conversely, the knockdown of all DVL genes using siRNA suppressed the expression of AXIN2 and c-Jun activated by Wnt3a stimulation (Fig. 1d). Exogenous expression of WNK1 rescued this suppression by DVL knock-down (Fig. 1d). These results indicate that WNK1 acts as a downstream element of Dvl in the Wnt signalling pathway.

### WNK regulates the Wnt signalling pathway by controlling the β-Catenin level.

We next analysed the relationship between WNK and β-Catenin in the Wnt signalling pathway. The expression of β-Catenin activated the target genes of the Wnt signalling pathway[6,7]. The knockdown of both WNK1 and WNK4 suppressed the expression of AXIN2 and c-Jun activated by β-Catenin expression (Fig. 1e). The knockdown of β-Catenin using siRNA suppressed the expression of AXIN2 and c-Jun activated by Wnt3a stimulation, while the exogenous expression of WNK1 could not rescue this suppression by β-Catenin knockdown (Fig. 1f). These results suggest that the function of β-Catenin in Wnt signalling is required, but not sufficient, for WNK expression. Therefore, we examined whether WNK regulates the β-Catenin levels. β-Catenin is degraded via the proteasome under normal conditions, and after Wnt stimulation, β-Catenin accumulates in the cytoplasm[6,7]. The knockdown of both WNK1 and WNK4 reduced the β-Catenin level under both normal conditions and Wnt-stimulated conditions (Fig. 1g). MG132, a proteasome inhibitor, blocked the degradation of β-Catenin under both conditions[29] (Fig. 1g). Moreover, MG132 still blocked the degradation of β-Catenin following the knockdown of both WNK1 and WNK4 (Fig. 1g). We also checked the ubiquitination level of β-Catenin. As shown in Fig. 1h, the expression of both WNK1 and WNK4 reduced the ubiquitination level of β-Catenin under MG132 treatment. These results suggest that WNK regulates the β-Catenin level by regulating the ubiqui-tination of β-Catenin.

### WNK is involved in the ubiquitination of β-Catenin via the GID complex.

β-Catenin is phosphorylated by GSK3β in the destruction complex and is ubiquitinated by βTrCP E3 ubiquitin ligase[11,30]. Because the N-terminal deletion of β-Catenin (β-Catenin ΔN) deletes the phosphorylation and ubiquitination sites, β-Catenin ΔN is no longer degraded and functions as a con-stitutively active form[31]. Because WNK is related to β-Catenin ubiquitination, we next examined the β-Catenin ΔN level under conditions with the knockdown of both WNK1 and WNK4. Surprisingly, similar to the normal β-Catenin level, the β-Catenin ΔN level was also reduced by the knockdown of both WNK1 and WNK4 (Fig. 2a). The levels of other constitutively active mutants of β-Catenin, such as β-Catenin S33Y and S45A, were also reduced (Fig. 2a). As shown above, the expression of WNK1 and WNK4 suppressed the ubiquitination of wild-type β-Catenin

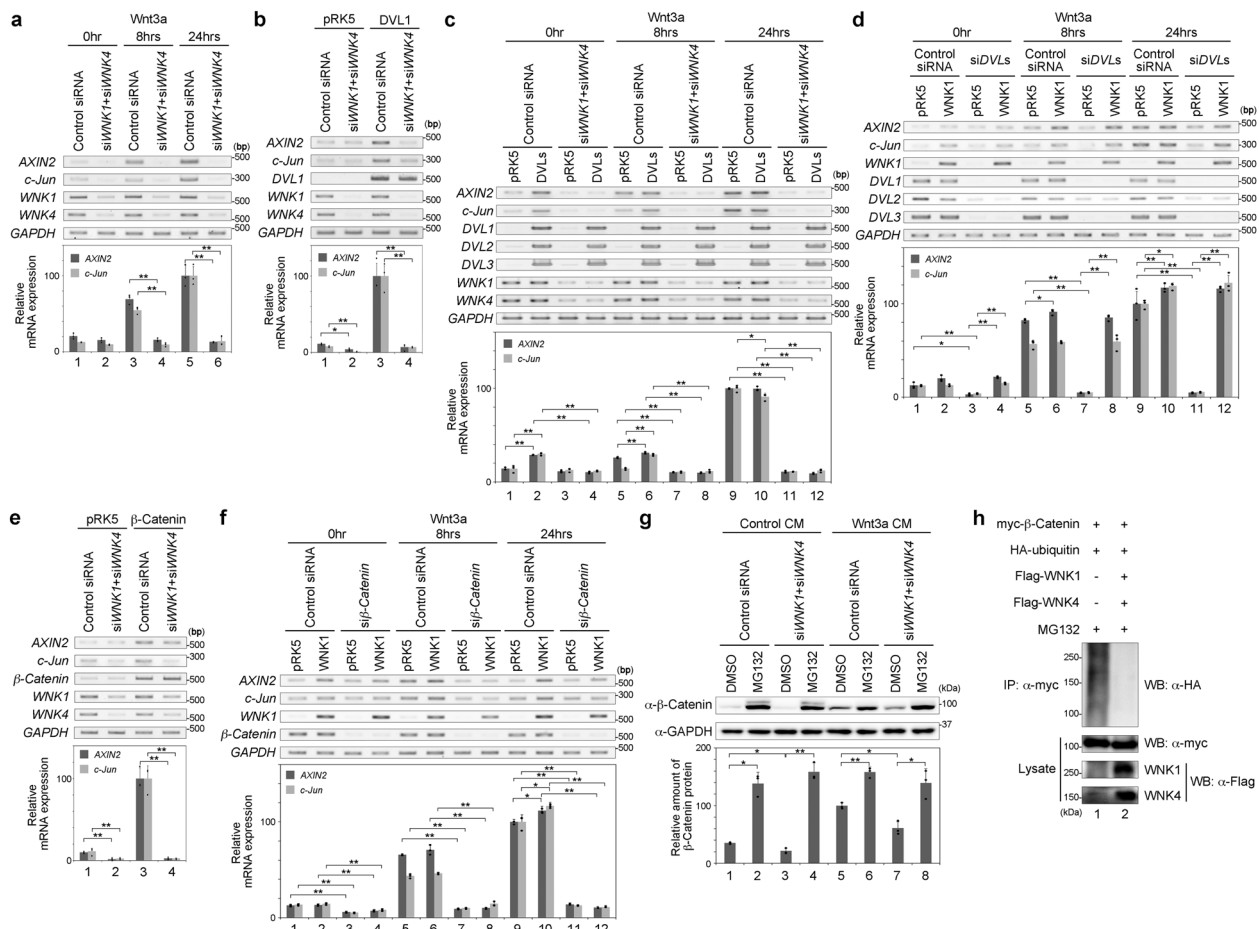

**Fig. 1 WNK is a positive regulator of the Wnt signalling pathway. a** Gene expression was examined by RT-PCR or quantitative RT-PCR in HEK293T cells following Wnt stimulation or the knockdown of both *WNK1* and *WNK4*. $n = 3$ biologically independent experiments. Dots indicate individual data. **b** Gene expression was examined by RT-PCR or quantitative RT-PCR in HEK293T cells following the expression of DVL1 or knockdown of both *WNK1* and *WNK4*. $n = 3$ biologically independent experiments. Dots indicate individual data. **c** Gene expression was examined by RT-PCR or quantitative RT-PCR in HEK293T cells following the treatment with Wnt3a stimulation, the expression of DVL1, DVL2 and DVL3, or knockdown of both *WNK1* and *WNK4*. $n = 3$ biologically independent experiments. Dots indicate individual data. **d** Gene expression was examined by RT-PCR or quantitative RT-PCR in HEK293T cells following Wnt stimulation of the expression of WNK1 or knockdown of *DVL1*, *DVL2* and *DVL3*. $n = 3$ biologically independent experiments. Dots indicate individual data. **e** Gene expression was examined by RT-PCR or quantitative RT-PCR in HEK293T cells following the expression of β-Catenin or knockdown of both *WNK1* and *WNK4*. $n = 3$ biologically independent experiments. Dots indicate individual data. **f** Gene expression was examined by RT-PCR or quantitative RT-PCR in HEK293T cells following Wnt stimulation, the expression of WNK1 or knockdown of β-Catenin. $n = 3$ biologically independent experiments. Dots indicate individual data. **g** Western blot analysis of endogenous protein expression following Wnt stimulation, the knockdown of both *WNK1* and *WNK4*, or MG132 (10 μM, 6 h) treatment in HEK293T cells. $n = 3$ biologically independent experiments. Dots indicate individual data. **h** Western blot analysis of ubiquitinated β-Catenin under the expression of WNK1 and WNK4 in HEK293T cells. Values and error bars express mean ± standard deviation (SD). * indicates $p < 0.05$ and ** indicates $p < 0.005$. $p$ value was calculated by Bonferroni test.

(Fig. 1g). Therefore, we checked the ubiquitination level of β-Catenin ΔN. Similar to wild-type β-Catenin, the ubiquitination level of β-Catenin ΔN was reduced by the expression of both WNK1 and WNK4 (Supplementary Fig. 1a). In the human colorectal cancer cell line SW480, which has a deletion in *APC*, β-Catenin is not degraded by the destruction complex[32]. In SW480 cells, the knockdown of both *WNK1* and *WNK4* led to a reduced β-Catenin level (Fig. 2b). Furthermore, the reduced β-Catenin level caused by the knockdown of both *WNK1* and *WNK4* could not be rescued by the knockdown of both *βTrCP* and *FBXW11* (also known as *βTrCP2*) in SW480 cells (Fig. 2c). These results suggest that WNK is not related to the ubiquitination of β-Catenin by the destruction complex containing βTrCP.

Previous reports identified several E3 ubiquitin ligases for β-Catenin other than βTrCP, such as SIAH1, Jade1, EDD, Mule, SHPRH and the GID complex (also known as the CTLH

complex)[11–17]. Among these ligases, Jade-1 interacts with the N-terminus of β-Catenin like βTrCP[12], and Mule targets β-Catenin only under activated Wnt conditions[17], suggesting that Jade-1 and Mule are unlikely to be related to WNK function. Furthermore, because EDD targets intact β-Catenin, EDD might not be involved in WNK function. Therefore, we examined whether the knockdown of the genes encoding SIAH1, SHPRH and GID complex could rescue the reduced β-Catenin level by the knockdown of *WNK*. We found that the reduction of β-Catenin by the knockdown of both *WNK1* and *WNK4* could not be rescued by the knockdown of *SIAH1* or *SHPRH* using siRNA in SW480 cells (Fig. 2d, e). These findings also suggest that SIAH1 or SHPRH is not involved in WNK function.

Our previous report suggests that the GID complex is involved in the ubiquitination of β-Catenin and is independent of the Wnt signalling pathway[16]. Among the components of the GID complex,

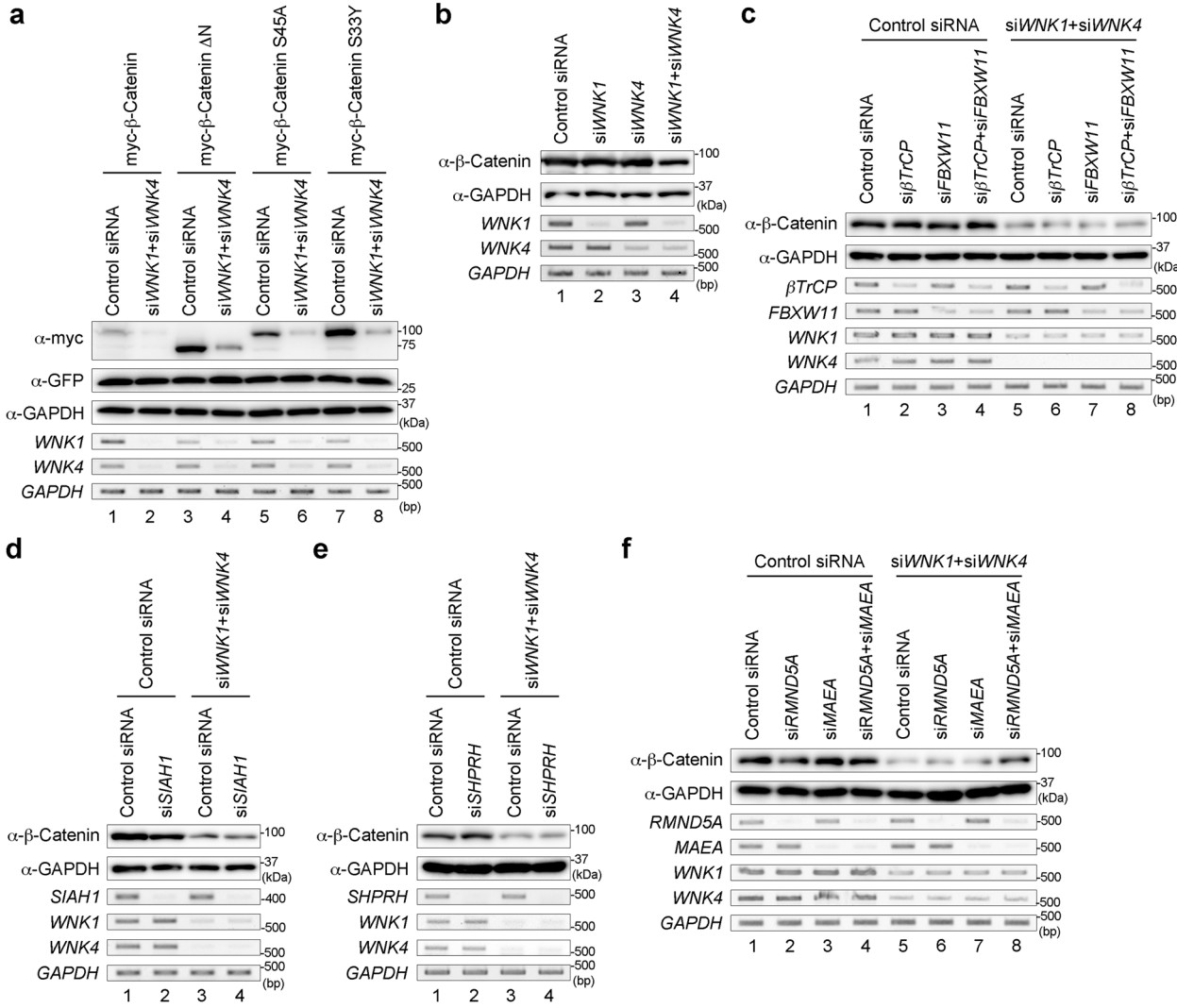

**Fig. 2 WNK regulates the protein level of β-Catenin through the GID complex. a** Western blot analysis of β-Catenin following the knockdown of both *WNK1* and *WNK4* in HEK293T cells. GFP is co-transfected for checking the transfection efficiency. **b** Western blot analysis of endogenous β-Catenin following the knockdown of *WNK1* and/or *WNK4* in SW480 cells. **c–f** Western blot analysis of endogenous β-Catenin following the knockdown of both *WNK1* and *WNK4*, or the knockdown of *βTrCP* and/or *βTrCP2* (**c**), *SIAH1* (**d**), *SHPRH* (**e**), or *RMND5A*, and/or *MAEA* (**f**) in SW480 cells.

*RMND5A* and *MAEA* (mammalian homologues of yeast *GID2* and *GID9*, respectively) encode E3 ubiquitin ligases[33–35]. We confirmed that RMND5A and MAEA ubiquitinated β-Catenin directly (Supplementary Fig. 1b). The reduction of β-Catenin by the knockdown of both *WNK1* and *WNK4* could be rescued by the knockdown of both *RMND5A* and *MAEA* (Fig. 2f). Next, we constructed deletion mutants of the RING finger domain of RMND5A and MAEA (RMND5A ΔR and MAEA ΔR, respectively) as dominant-negative forms. The reduction of β-Catenin by the knockdown of both *WNK1* and *WNK4* could also be rescued by the expression of both RMND5A ΔR and MAEA ΔR (Supplementary Fig. 1c). These results suggest that the GID complex is involved in the ubiquitination of β-Catenin associated with WNK function.

**WNK attenuates the interaction between β-Catenin and MAEA.** As shown above, because RMND5A and MAEA of the GID complex are related to WNK, we examined the interaction between WNK and RMND5A or MAEA by immunoprecipitation. We transiently expressed Flag-tagged RMND5A or MAEA together with myc-tagged WNK1 or WNK4. When the cell extracts were subjected to immunoprecipitation with anti-myc antibodies, followed by immunoblotting, we found that RMND5A interacted

with WNK4 (Fig. 3a) and that MAEA interacted with WNK1 and WNK4 (Fig. 3b). Because RMND5A and MAEA interacted with each other, like yeast GID2 and GID9[36] (Fig. 3c), it was suggested that WNK forms a complex with the GID complex.

MAEA interacted with β-Catenin, but RMND5A was not detected (Fig. 3d, e). Therefore, we examined whether WNK affected the interaction between β-Catenin and MAEA. The interaction between β-Catenin and MAEA was reduced by the expression of both WNK1 and WNK4, but was not reduced by the expression of only WNK1 or WNK4 (Fig. 3d). By contrast, the knockdown of *WNK1* and/or *WNK4* increased the interaction of β-Catenin and MAEA (Fig. 3e). We next examined whether WNK or the GID complex has the opposite effect on the ubiquitination of β-Catenin. As shown in Fig. 3f, the expression of both RMND5A and MAEA increased the ubiquitination of β-Catenin and rescued the reduction of β-Catenin ubiquitination by WNK expression. These results suggest that the presence of WNK attenuates the interaction between β-Catenin and MAEA and inhibits the ubiquitination of β-Catenin by the GID complex.

**The WNK inhibitor functions as an inhibitor of the WNK–GID complex interaction.** In WNK signalling, SPAK/OSR1 are cofactors

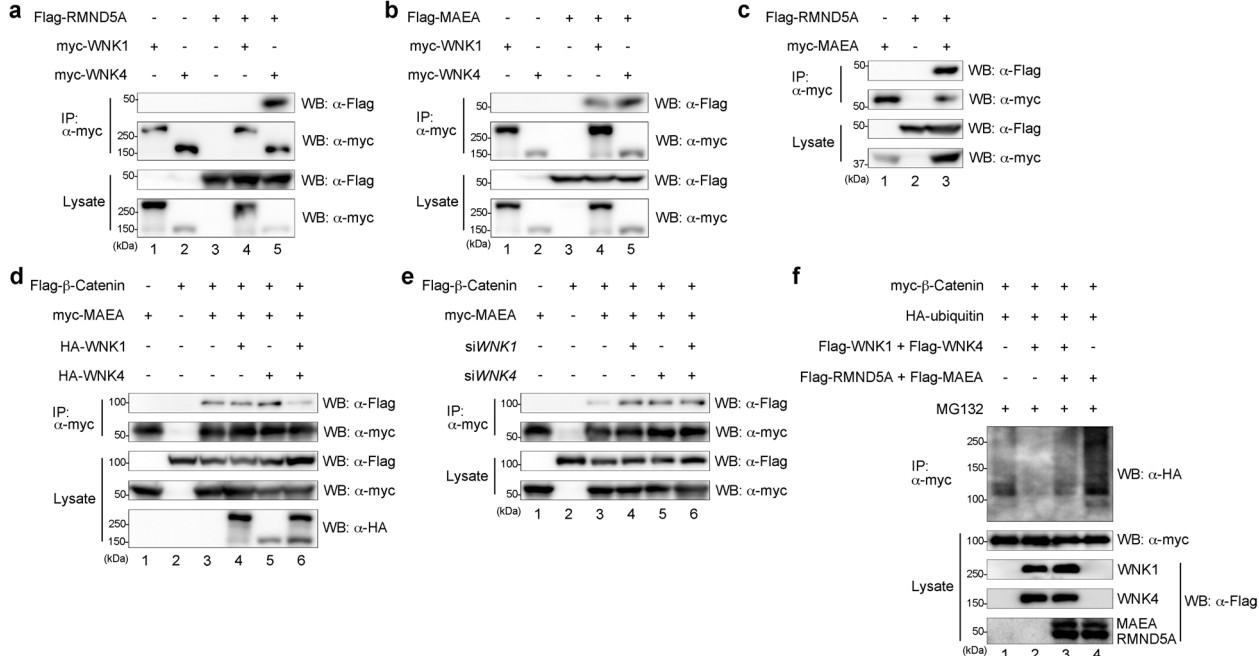

**Fig. 3 WNK attenuates the interaction between β-Catenin and MAEA. a, b** The interaction between WNK1 or WNK4, and RMND5A (**a**) or MAEA (**b**) was examined in HEK293T cells by co-immunoprecipitation. **c** The interaction between RMND5A and MAEA was examined in HEK293T cells by co-immunoprecipitation. **d, e** The interaction between β-Catenin and MAEA following the expression of WNK1 and/or WNK4 (**d**) or the knockdown of *WNK1* and/or *WNK4* (**e**) was examined in HEK293T cells by co-immunoprecipitation. **f** Western blot analysis of ubiquitinated β-Catenin following the expression of WNK1 and WNK4, and/or RMND5A and MAEA in HEK293T cells.

and substrates of WNK. Previous research identified two compounds as inhibitors of the WNK signalling pathway, which inhibit the WNK–OSR interaction[37]. Thus, we checked whether these inhibitors also reduced the β-Catenin level. In SW480 cells, we found that STOCKS2S-26016 (hereafter referred to as 26016) reduced the β-Catenin level, but STOCK1S-50699 (hereafter referred to as 50699) did not (Fig. 4a). The expression levels of *AXIN2* and *c-Jun* activated by Wnt3a stimulation in HEK293T cells were also inhibited in the presence of 26016 (Fig. 4b). We next examined the function of 26016 in WNK signalling regarding β-Catenin degradation. We found that 26016 inhibited the interaction of WNK1 and WNK4 (Fig. 4c) and that the interaction between β-Catenin and MAEA increased under 26016 treatment (Fig. 4d), similar to the results of WNK knockdown (Fig. 3e). Furthermore, treatment with 26016 increased the ubiquitination level of β-Catenin (Fig. 4e). These results suggest that the WNK inhibitor 26016 functions as an inhibitor of the Wnt signalling pathway, similar to the effect of WNK knockdown.

Derivatives of 26016 were developed in a previous study[38]. Thus, we checked whether derivatives of 26016 also affected the β-Catenin level in SW480 cells and found that derivative #13 (hereafter referred to as #13) also reduced the β-Catenin level (Fig. 4f). #13 also inhibited the interaction between WNK1 and WNK4 (Fig. 4c), increased the interaction between β-Catenin and MAEA (Fig. 4d), and suppressed the expression levels of *AXIN2* and *c-Jun* activated by Wnt3a stimulation (Fig. 4g). Moreover, #13 increased the ubiquitination level of β-Catenin (Fig. 4e), suggesting that 26016 and #13 share similar activities for WNK signalling and β-Catenin degradation. Furthermore, we found that the knockdown of RMND5A and MAEA could rescue the reduced β-Catenin level by the effect of 26016 or #13 (Fig. 4h, i, respectively). The expression of the dominant-negative forms of RMND5A and MAEA (RMND5A ΔR and MAEA ΔR) could also rescue the reduced β-Catenin level (Supplementary Fig. 2a, b, respectively). We next examined the effect of WNK inhibitors on

WNK and GID complex. We found that WNK inhibitors suppressed the binding between WNK1 and MAEA, or WNK4 and MAEA (Fig. 4j, k, respectively), but not the binding between WNK4 and RMND5A (Supplementary Fig. 2c). These results suggest that WNK inhibitors 26016 and #13 work as inhibitors of the WNK–GID complex interaction.

**WNK can regulate the β-Catenin level in colorectal cancer cells.**
In colorectal cancer cells, missense mutations are frequently found in exon 3 of *CTNNB1*, which encodes the N-terminal region of β-Catenin, including serine–threonine phosphorylation sites for GSK3β that induce β-Catenin degradation[39]. Mutations of the N-terminal region such as S33, S37 and S41 cause stabilization of β-Catenin and activation of target gene expression driving tumour formation[40]. As shown above, we identified that WNK regulates the β-Catenin level through GID E3 ligase and showed that the N-terminal region of β-Catenin is dispensable for WNK-dependent degradation (Figs. 2 and 3), suggesting that the mechanism of WNK-dependent β-Catenin degradation is a candidate therapeutic target for colorectal cancer. To explore this possibility, we checked the effect of WNK on the β-Catenin level in several colorectal cancer cell lines (DLD1 and HCT116 cells) in addition to SW480 cells. Consistent with the data from SW480 cells, knockdown of both WNK1 and WNK4 reduced the β-Catenin levels in DLD1 and HCT116 cells (Fig. 5a and Supplementary Fig. S3a, respectively). Next, we checked the effect of the RMND5A ΔR and MAEA ΔR mutants on the β-Catenin level in these cells. The reduction in β-Catenin level by the knockdown of both WNK1 and WNK4 was rescued by the expression of the dominant-negative forms of RNMD5A and MAEA (Fig. 5b and Supplementary Fig. S3b, respectively). In addition, the reduction of β-Catenin by WNK inhibitors 26016 and #13 was also restored by RMND5A ΔR and MAEA ΔR in these cells (Fig. 5c, d and Supplementary Fig. S3c, d, respectively). These results suggest

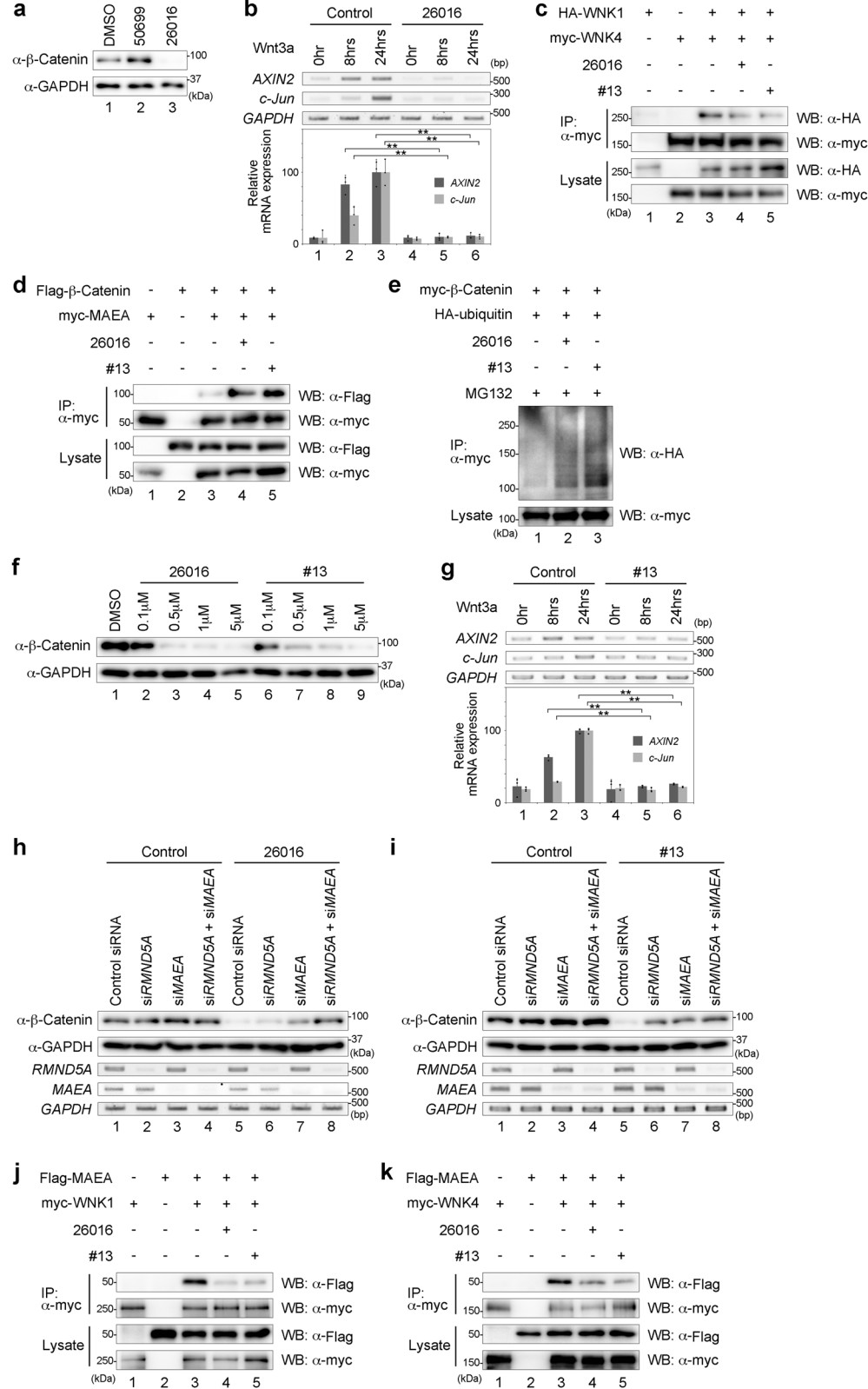

that WNK regulates the β-Catenin level in various colorectal cancer cells.

## WNK inhibitors induce xenograft tumour growth regression.

Previous studies reported that the induction of β-Catenin degradation by small molecules is effective to suppress colorectal cancer development[41], suggesting that β-Catenin degradation mediated by the WNK inhibitors 26016 and #13 would repress colorectal cancer development. Therefore, we evaluated the anti-tumour effect of WNK inhibitors. SW480 cells were treated with 26016 and #13, and cell viability or growth was measured. 26016 effectively induced cell death after 24 h, while #13 only slightly affected cell death, even at a high dose (Fig. 6a). However, #13 caused

**Fig. 4 WNK inhibitors function as Wnt inhibitors. a** Western blot analysis of endogenous β-Catenin following treatment with 26016 in SW480 cells. **b** Gene expression was examined by RT-PCR or quantitative RT-PCR in HEK293T cells following Wnt stimulation or treatment with 26016. $n = 3$ biologically independent experiments. Dots indicate individual data. **c** The interaction between WNK1 and WNK4 following treatment with 26016 or #13 was examined in HEK293T cells by co-immunoprecipitation. **d** The interaction between β-Catenin and MAEA following treatment with 26016 or #13 was examined in HEK293T cells by co-immunoprecipitation. **e** Western blot analysis of ubiquitinated β-Catenin following treatment with 26016 or #13 in HEK293T cells. **f** Western blot analysis of endogenous β-Catenin following treatment with 26016 or #13. **g** Gene expression by RT-PCR or quantitative RT-PCR analysis was examined in HEK293T cells following Wnt stimulation or treatment with #13. $n = 3$ biologically independent experiments. Dots indicate individual data. **h, i** Western blot analysis of endogenous β-Catenin following the knockdown of *RMND5A* and/or *MAEA*, or treatment with 26016 (**h**) or #13 (**i**) in SW480 cells. **j** The interaction between WNK1 and MAEA following treatment with 26016 or #13 was examined in HEK293T cells by co-immunoprecipitation. **k** The interaction between WNK4 and MAEA following treatment with 26016 or #13 was examined in HEK293T cells by co-immunoprecipitation. Values and error bars express mean ± standard deviation (SD). ** indicates $p < 0.005$. $p$ value was calculated by Bonferroni test.

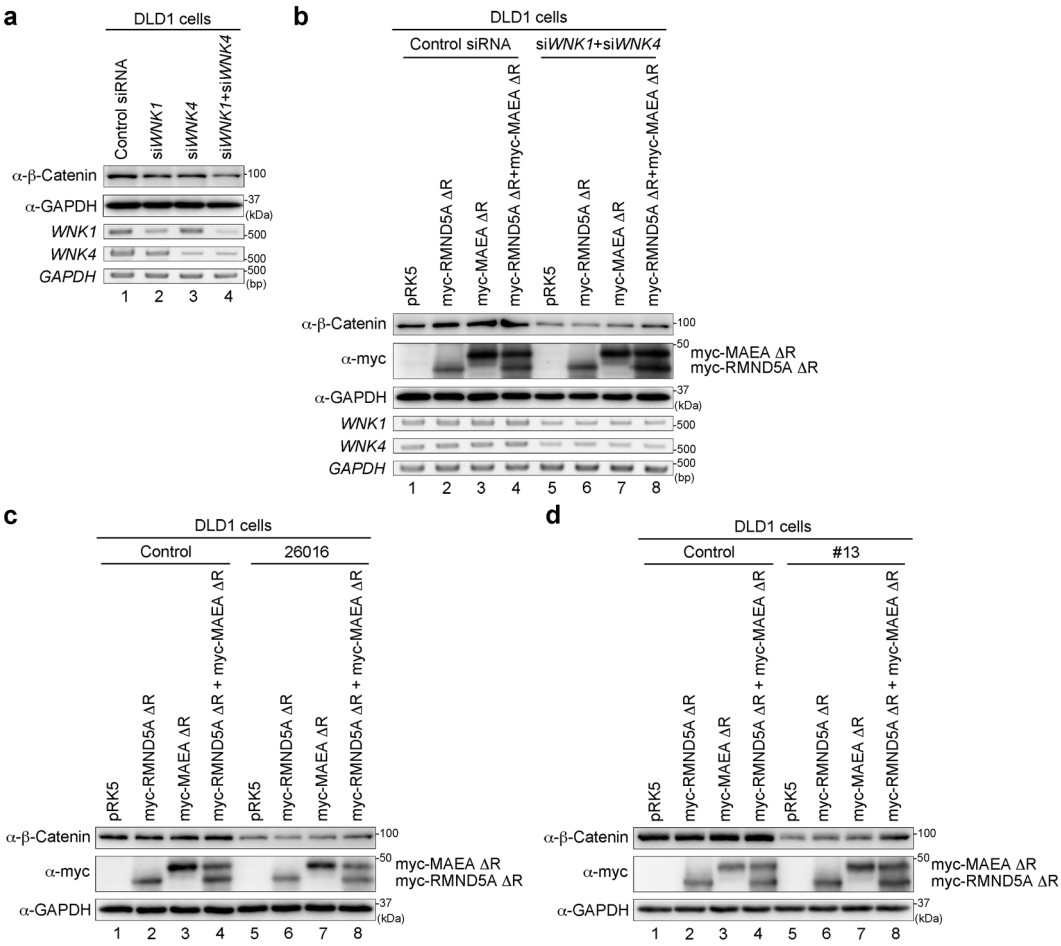

**Fig. 5 WNK regulates the protein level of β-Catenin in DLD1 colorectal cancer cells. a** Western blot analysis of endogenous β-Catenin following the knockdown of *WNK1* and/or *WNK4* in DLD1 cells. **b–d** Western blot analysis of endogenous β-Catenin following the knockdown of both *WNK1* and *WNK4* (**b**), treatment with the WNK inhibitor 26016 (**c**) or #13 (**d**), or the expression of RMND5A ΔR and/or MAEA ΔR in DLD1 colorectal cancer cells.

significant suppression of cell growth (Fig. 6b). We next analysed the effects of 26016 and #13 on xenograft tumour formation in vivo. SW480 cells were subcutaneously injected into immuno-deficient mice, 26016 or #13 was injected intraperitoneally twice per week, and tumour growth was monitored. We observed little or no effect of 26016 on xenograft progression at low doses, but at a high dose of 26016, four of five mice died after only two injections (Fig. 6c). This suggests that 26016 has more severe toxicity than efficacy in vivo. By contrast, treatment of #13 resulted in reduced tumour size and weight in a dose-dependent manner compared with the findings for tumours from vehicle injection (Fig. 6d, e). Moreover, the amount of β-Catenin in the tumours treated with #13 was reduced in a manner dependent on the concentration of #13 and tumour size (Fig. 6f). These findings

suggest that #13 has low toxicity and prevents colorectal cancer development through β-Catenin degradation.

## Discussion

In the Wnt signalling pathway, to regulate β-Catenin expression, the destruction complex phosphorylates β-Catenin and then ubiquitinates it through βTrCP[7]. The ubiquitinated β-Catenin is degraded by the proteasome. However, several E3 ubiquitin ligases other than βTrCP have been identified, including the GID complex[11–17]. In this study, we showed that WNK affected the protein and ubiquitination levels of not only wild-type β-Catenin (Fig. 1) but also β-Catenin ΔN (Fig. 2). We also found that WNK can control the β-Catenin level through the GID complex. Knockdown of RMND5A and MAEA of the GID complex could

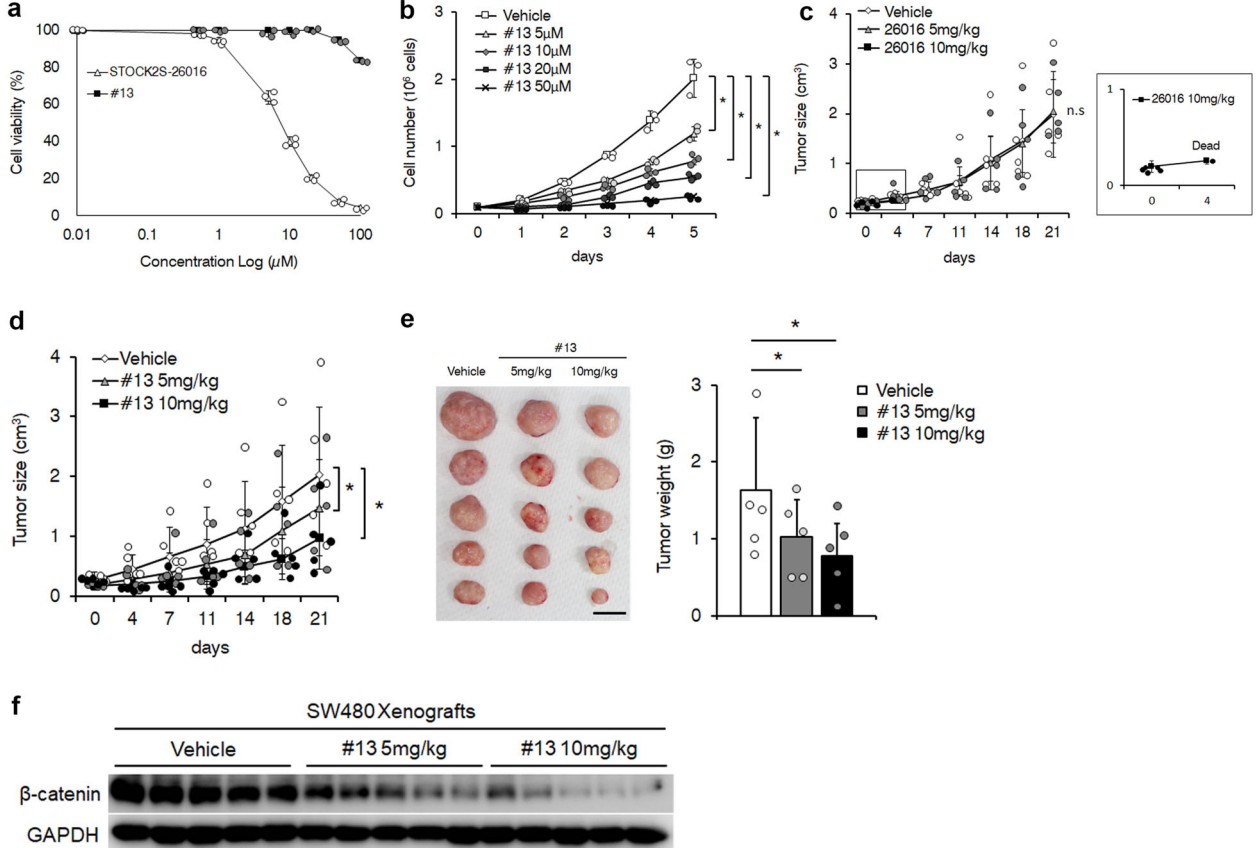

**Fig. 6 WNK inhibitor suppresses the xenograft development of colorectal cancer cells. a** Cell viability assays of SW480 cells treated with 26016 or #13 for 24 h. $n = 3$ biologically independent experiments. Circles indicate individual data. **b** Growth curves of SW480 cells treated with #13 for the indicated days. $n = 3$ biologically independent experiments. Circles indicate individual data. **c**, **d** Immunodeficient mice were subcutaneously injected with SW480 cells ($1 \times 10^6$ cells). The dose of 5 or 10 mg/kg of 26016 (**c**) and #13 (**d**) was intraperitoneally injected twice a week for 3 weeks, and the tumour size was measured before the injection of inhibitors. The inset shows high magnification of the rectangle in (**c**). $n = 5$ biologically independent animals. Circles indicate individual data, n.s. indicates not significant. **e** The tumours and tumour weights after injection of #13 for 3 weeks in each mouse are indicated in (**d**). Circles indicate individual data. Scale bar, 1 cm. **f** Western blot analysis of β-Catenin expression level in SW480 xenografts after injection of #13. Values and error bars express mean ± standard deviation (SD). * indicates $p < 0.05$. $p$ value was calculated by Student's $t$ test versus vehicle.

rescue the effect of WNK knockdown (Fig. 2), and the presence of both WNK1 and WNK4 could change the interaction between β-Catenin and MAEA (Fig. 3), suggesting that the simultaneous stabilization of WNK1 and WNK4 would be important in the destruction complex of the canonical Wnt signalling pathway. The levels of both WNK1 and WNK4 were increased by Wnt stimulation (Supplementary Fig. S4), as well as the β-Catenin level, although the transcription level of neither WNK1 nor WNK4 changed. Moreover, we previously reported that GSK3β is bound to WNK1[27], suggesting that WNK proteins might be located in the destruction complex of Wnt signalling. Taken together, our data showed that the presence of both WNK1 and WNK4 might provide fine-tuning to attenuate the interaction between β-Catenin and the GID complex and inhibit the ubiquitination of β-Catenin in the Wnt signalling pathway.

A previous study reported that WNK is involved in the hyperphosphorylation of Dsh in *Drosophila* S2 cells and that WNK functions as a positive regulator of the Wnt signalling pathway[28]. However, we found that the knockdown of WNK1 and WNK4 had no effect on DVL1 phosphorylation induced by Wnt stimulation (Supplementary Fig. 5), and that WNK acts downstream of DVL (Fig. 1). Furthermore, WNK affected the β-Catenin ΔN level (Fig. 2) and was not associated with βTrCP E3 ubiquitin ligase (Fig. 3). Therefore, our results suggest that DVL is not related to WNK function in the Wnt signalling pathway.

Further study will be required concerning the interaction between WNK and DVL.

β-Catenin degradation under normal conditions is likely regulated by phosphorylation through the destruction complex of the Wnt signalling pathway and ubiquitination by βTrCP. In addition, several E3 ubiquitin ligases other than βTrCP, such as SIAH1, SHPRH and GID complex[11,13,15,16], are involved in β-Catenin degradation, indicating that β-Catenin is degraded through multiple mechanisms. We found that the β-Catenin degradation mediated through the knockdown of WNK could not be rescued by knockdown of βTrCPs, SIAH1 or SHPRH (Fig. 2). These results indicate that WNK controls β-Catenin degradation independently of βTrCPs, SIAH1 or SHPRH. However, WNK might be located near the destruction complex as we discussed above, suggesting that the GID complex localizes near the destruction complex with βTrCP. Furthermore, we found that the protein levels of WNK1 and WNK4 were upregulated by Wnt stimulation (Supplementary Fig. S4). This suggests the possibility that Wnt signalling also regulates the activity of the GID complex for fine-tuning the level of β-Catenin protein with βTrCP-mediated degradation.

In this study, we found that the knockdown of WNK caused the reduction of not only the wild type but also the constitutively active type of β-Catenin through the GID complex (Fig. 2). This suggests that WNK ordinarily prevents the interaction between

the E3-ligases RMND5A and MAEA, and β-Catenin. A previous paper showed that SHPRH is a negative regulator of Wnt signalling and is involved in the degradation of both the wild type and active forms of nuclear β-Catenin[15]. However, we did not detect the involvement of SHPRH in the regulation of β-Catenin degradation by WNK (Fig. 2). Furthermore, the lysine demethylase KDM2a/b was recently identified as a regulator of β-Catenin methylation[42]. KDM2a/b in the nucleus demethylates β-Catenin, which in turn enhances the ubiquitination level of β-Catenin and degradation of β-Catenin, even β-Catenin ΔN. Because WNK in the destruction complex protects the GID complex activity in the cytoplasm, it appears that WNK functions independently of SHPRH and KDM2a/b.

As stated above, we identified that WNK controls the β-Catenin level via the GID complex (Figs. 2 and 3). Moreover, we found that the WNK inhibitors 26016 and #13 function as inhibitors of the Wnt signalling pathway. As shown in Figs. 4 and 5 and Supplementary Fig. 3, WNK inhibitors reduced the β-Catenin level in several colorectal cancer cell lines. However, the WNK inhibitor 26016 caused death in mice, and the other inhibitor, #13, clearly repressed the growth of xenograft tumour cells (Fig. 5). The structural difference between 26016 and #13 is a substitution of the side chain from 2-furanylmethylamino to 3-cyanobenzamido[38]. We still do not know how this structural difference affects the viability of mice, but our results indicate that the WNK inhibitor #13 likely works as an antitumour drug in vivo. Because the regulation of β-Catenin degradation by WNK is so unique, the WNK inhibitor is a potential anti-tumour drug. β-Catenin is involved not only in colorectal cancer but also in other cancers, such as breast cancer, lung cancer and leukaemia[6]. Furthermore, abnormal β-Catenin accumulation causes other diseases, such as autosomal dominant polycystic kidney disease[5]. These findings indicate that WNK is a potential target for treating cancers or other diseases via the dysregulation of β-Catenin.

## Methods

**Ethics statement**. All animal experiments were performed in accordance with the ethical guidelines of Tokyo Medical and Dental University, Nippon Medical School and the Law and Notification of the Government of Japan. Animal protocols were reviewed and approved by the animal welfare committee of Tokyo Medical and Dental University and the Ethics Committee on Animal Experiments of Nippon Medical School.

**Cultured cell lines**. The cell lines used in this study were HEK293T, SW480, HCT116 and DLD1. Dulbecco's Modified Eagle Medium (DMEM) supplemented with 10% foetal bovine serum (FBS) was used to culture HEK293 and SW480 cells, McCoy's 5A supplemented with 10% FBS was used to culture HCT116 cells and RPMI 1640 supplemented with 10% FBS was used to culture DLD1 cells. For the co-transfection of short interfering RNA (siRNA) and plasmids, we used the *TransIT*-X2 Dynamic Delivery System (Mirus Bio) and polyethyleneimine (Polysciences), respectively. The siRNA target sequences are summarized in Supplementary Table 1.

**Antibodies and chemicals**. The antibodies used in this study were as follows: rabbit anti-β-Catenin (Abcam), rabbit anti-Flag, rabbit anti-HA, rabbit anti-myc (Medical & Biological Laboratory), rabbit anti-WNK1, rabbit anti-WNK4, mouse anti-myc (Cell Signaling Technology) and anti-rabbit HRP-conjugated (GE) antibodies.

STOCKS2S-26016 was purchased from Tocris and #13 was synthesized at the Kagechika laboratory. MG132 was purchased from Merck.

**Reverse-transcription polymerase chain reaction (RT-PCR) analysis**. Total RNA was isolated using TRIzol reagent (Invitrogen). cDNA synthesis was carried out using Moloney murine leukaemia virus reverse transcriptase (Invitrogen). *GAPDH* was used to normalize the cDNA samples. The sequences of the PCR primer pairs are summarized in Supplementary Table 2.

**Quantification**. Quantitative PCR was performed using the Applied Biosystems 7300 Real-Time *PCR* Cycler (ABI) and THUNDERBIRD SYBR qPCR Mix (TOYOBO). *GAPDH* was used to normalize the cDNA samples.

**Plasmids**. We made several WNK1 or WNK4 constructs with different molecular tags (myc, HA) from WNK1 or WNK4 constructs with Flag-tag[26]. *DVL1*, *DVL2*, *DVL3*, *β-Catenin*, *RMND5A* and *MAEA* cDNA were obtained by RT-PCR and were constructed in the pRK5 vector. To construct the constitutively active form of β-Catenin, as well as dominant-negative forms of RMND5A and MAEA, we performed site-directed mutagenesis.

**Immunoprecipitation**. HEK293T cells were transfected with the indicated expression vector. Lysates were prepared from transfected cells and were immunoprecipitated with the indicated antibodies and Protein A/G PLUS-agarose (Santa Cruz Biotechnology). Immunoprecipitates were subjected to sodium dodecyl sulphate–polyacrylamide gel electrophoresis and western blotting, and were then detected using the image analyser LAS-4000 Mini (GE).

**Cell viability and growth**. Cell viability was analysed using WST1 assays (Takara). SW480 cells ($2 \times 10^4$ cells) in 96-well plates were treated with the indicated chemicals for 24 h, and WST1 was added to each well. The absorbance (450 or 630 nm) was measured using a microplate reader, and the viability of WNK inhibitor-treated cells was determined. For cell growth, SW480 cells ($1 \times 10^5$) were seeded in six-well plates and were treated with WNK inhibitor #13. The number of #13-treated cells was counted by Vi-CELL (Beckman) on the indicated days, and cell growth curves were created.

**Xenograft mouse model**. Five-week-old BALB/cAJ1-*nu/nu* male mice were obtained from CLEA Japan. All of the mice used in the experiments were assigned randomly. SW480 cells ($1 \times 10^6$) were subcutaneously injected into the mice. When the tumour size reached approximately 200 mm$^3$, all of the mice were injected intraperitoneally with the indicated chemicals twice weekly for 3 weeks at a dose of 5 or 10 mg/kg. The tumour sizes were measured before each injection. At the end of the experiments after injection for 3 weeks, the animals were sacrificed and the tumour weights were measured. Collected tumours were homogenized and applied for immunoblotting assays to analyse β-Catenin expression.

**Statistics and reproducibility**. All experiments were conducted at least three times independently. The data were analysed using Microsoft Excel (Microsoft) and StatPlus (AnalystSoft). Values and error bars express mean ± standard deviation (SD) and are representative of at least three independent experiments.

**Reporting summary**. Further information on the research design is available in the Nature Research Reporting Summary linked to this article.

## Data availability

The material used in this study as #13 is available from H.K. upon reasonable request. The other data are available from the corresponding author upon request. Source data behind the graphs are available in Supplementary Data 1. All full immunoblot and gel images are shown in Supplementary Fig. 6.

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

## Acknowledgements

We thank Yoko Mitsutomo for technical assistance. This study was supported by Grants-in-Aid for Scientific Research from the Ministry of Education, Culture, Sports, Science and Technology of Japan (A.S., M.S. and H.S.) and Nanken-Kyoten, Tokyo Medical and Dental University (H.S.). We thank Edanz Group (https://en-author-services.edanzgroup.com/ac) for editing a draft of this manuscript.

## Author contributions

A.S. and H.S. designed the study. A.S., M.S. and T.G. performed experiments and analysed the data. M.S. and N.T. performed xenograft experiments. H.M. and H.K. synthesized chemical compounds. All authors discussed the data. A.S., M.S. and H.S. wrote the manuscript.

## Competing interests

The authors declare no competing interests.
