## [Peer Review File · Communications Biology]

Reviewers' comments:

Reviewer #1 (Remarks to the Author):

Sato et al present well-researched data that WNK is a regulator of b-catenin degradation by the GID complex, which follows their previously published reports that GID is involved in CTNNB1 ubiquitination.

One issue is how significant these observations are in vivo (in animal models other than the xenografts presented here) but I understand this is a different line of experiments.

The only significant criticism I have is with regards to the Dvl data which are not well developed:

1. There is no confirmation of DVL1 phosphorylation by WNK
2. Since there is no data to suggest that Dvl is involved in WNK, it is not clear to this Reviewer what the significance of this report is.

Other minor critiques:

1. Figures 3F and 4E are unclear
2. None of the figures has significance indicated in the graphs - this becomes difficult to follow, especially since in several experiments the error bar is too high

Reviewer #2 (Remarks to the Author):

In this manuscript, the authors report that WNK1/4 functions as a positive regulator of Wnt signaling by inhibiting the degradation of b-catenin, a key transducer of the Wnt signaling pathway. The E3 ubiquitin ligases RMND5A and MAEA, which are components of the GID complex, ubiquitinate and degrade β -catenin independently of the GSK3 β - β TrCP pathway. The authors show that WNK1/4 interacts with RMND5A and MAEA to reduce the interaction between MAEA and β -catenin, and thus leading to inhibition of β -catenin degradation by MAEA. They also show that WNK inhibitors promote the degradation of β -catenin through RMND5A and MAEA and suppress the growth of xenograft tumor cells.

Overall, the findings are novel and interesting for publication in Communications Biology. However, the following concerns need to be addressed before publication.

(1) Fig. 1C. When the authors transiently transfected WNK1 into DVLs-depleted cells, WNK1 completely rescued the defects in AXIN2 and c-Jun expression caused by DVLs knockdown. However, transfection efficiency is not 100%. The authors should explain this point.

(2) Fig. 1F. This reviewer cannot see any difference in the amount of β -catenin between lanes 1 and 3 or lanes 5 and 7. The authors should present these data quantitatively.

(3) Fig. 2A. This reviewer speculates that the authors transiently transfected myc- β -catenin into HEK293 cells. If so, it is difficult to conclude that the down-regulation of β -catenin is due to

degradation. There remains the possibility that the transfection efficiency or protein translation efficiency might have changed. The authors should use stable cell lines or co-express unaffected proteins such as GFP to verify equal transfection efficiency.

(4) In page 13 of the “Results” section, the authors mention that “WNK inhibitors 26016 and #13 function as inhibitors of the WNK-GID complex interaction”. However, there is no evidence to support this possibility. The authors should examine the effects of WNK inhibitors on WNK-GID complex interaction.

(5) Based on the results of xenograft tumor cells treated with WNK inhibitor #13, the authors argue that WNK might be valuable as a potential therapeutic target for cancer via dysregulation of β -catenin. However, there is no evidence that #13 suppresses the growth of xenograft tumor cells through β -catenin degradation. At least, the authors should compare the amount of β -catenin in tumors between vehicle-injected and #13-injected mice.

(6) It is of great interest that the WNK-GID pathway causes β -catenin degradation independently of β TrCP. However, the physiological relationship between the GID complex and β TrCP in β -catenin degradation remains unclear. The authors should discuss this point.

Dear REVIEWER 1,

As you have suggested, we have made a number of significant changes to our manuscript and feel that these changes have substantially improved the quality of the paper. We have addressed your concerns point-by-point as follows.

The only significant criticism I have is with regards to the Dvl data which are not well developed:

1. There is no confirmation of DVL1 phosphorylation by WNK

In accordance with this comment, we confirmed DVL1 phosphorylation using Phos-tag containing acrylamide gel. We detected that Wnt stimulation induced DVL1 phosphorylation, but the knockdown of *WNK1* and *WNK4* had no effect on the phosphorylation levels of DVL1 (Supplementary Figure 5). This result suggest that WNK is not involved in DVL1 phosphorylation by Wnt stimulation. We have also added this result in Discussion.

2. Since there is no data to suggest that Dvl is involved in WNK, it is not clear to this Reviewer what the significance of this report is.

In this paper, we focused on the mechanism that WNK regulates the degradation of β -catenin. On the other hand, we found that the knockdown of both *WNK1* and *WNK4* suppressed the expression of Wnt target genes (*AXIN2* and *c-Jun*) activated by DVL1 expression (Fig. 1B), and that the exogenous expression of WNK1 rescued the suppression of Wnt target gene expression by *DVLs* knockdown (Fig. 1D). In addition, we have also obtained new data that the exogenous expression of all DVL genes (DVL1, DVL2 and DVL3) did not rescue the suppression of Wnt induced gene expression by the knockdown of *WNK1* and *WNK4* (Fig. 1C). Taken together, these data indicate that WNK1 acts as a downstream element of Dvl. Thus, we believe that DVL is not directly involved in the degradation of β -catenin by WNK function in the Wnt signaling pathway.

Other minor critiques:

1. Figures 3F and 4E are unclear

As pointed by the reviewer, we have improved the results (Figure 3F and 4E).

2. None of the figures has significance indicated in the graphs - this becomes difficult to follow, especially since in several experiments the error bar is too high

As suggested by the reviewer, we have added the statistical analysis to all graphs. Especially, Figure 4G was improved since we mis-calculated.

Thanks again for your constructive criticisms.

Sincerely,

Hiroshi Shibuya

Dear REVIEWER 2,

As you have suggested, we have made a number of significant changes to our manuscript and feel that these changes have substantially improved the quality of the paper. We have addressed your concerns point-by-point as follows.

(1) Fig. 1C. When the authors transiently transfected WNK1 into DVLs-depleted cells, WNK1 completely rescued the defects in AXIN2 and c-Jun expression caused by DVLs knockdown. However, transfection efficiency is not 100%. The authors should explain this point.

As suggested by the reviewer, we agree that the transfection efficiency is not 100%. We checked the transfection efficiency in our methods using GFP, and confirmed that the efficiency is at least 70% as shown below. Since the transfection efficiency is very high and the experiments were performed by co-transfection both with siRNAs and the expression plasmids, we believe that the results of the rescue experiments are not artificial.

Brightfield

GFP

(2) Fig. 1F. This reviewer cannot see any difference in the amount of β -catenin between lanes 1 and 3 or lanes 5 and 7. The authors should present these data quantitatively.

As suggested by the reviewer, we quantified the amount of β -Catenin, and add the statistical data to Fig. 1F.

(3) Fig. 2A. This reviewer speculates that the authors transiently transfected myc- β -catenin into HEK293 cells. If so, it is difficult to conclude that the down-regulation of β -catenin is due to degradation. There remains the possibility that the transfection efficiency or protein translation efficiency might have changed. The authors should use stable cell lines or co-express unaffected proteins such as GFP to verify equal transfection efficiency.

As recommended by the reviewer, we performed the experiments by co-transfection with GFP, and confirmed that the transfection efficiency and translation efficiency in our experiments were not changed. We have improved the data set of Fig. 2A and the legend.

(4) In page 13 of the “Results” section, the authors mention that “WNK inhibitors 26016 and #13 function as inhibitors of the WNK-GID complex interaction”. However, there is no evidence to support this possibility. The authors should examine the effects of WNK inhibitors on WNK-GID complex interaction.

As recommended by the reviewer, we examined the effect of WNK inhibitors to WNK and GID complex. WNK inhibitors suppressed the binding between WNK1 and MAEA, and WNK4 and MAEA, but not WNK4 and RMND5A, suggesting that WNK inhibitors worked as inhibitors of WNK-GID complex interaction. We have added the data (Figure 4J, 4K and Supplementary Figure 2C), and improved the manuscript.

(5) Based on the results of xenograft tumor cells treated with WNK inhibitor #13, the authors argue that WNK might be valuable as a potential therapeutic target for cancer via dysregulation of β -catenin. However, there is no evidence that #13 suppresses the growth of xenograft tumor cells through β -catenin degradation. At least, the authors should compare the amount of β -catenin in tumors between vehicle-injected and #13-injected mice.

As recommended by the reviewer, we performed the Western blotting analysis of β -Catenin in Xenograft tumors. The amount of β -Catenin in Xenograft tumors treated

with #13 were reduced depending on drug concentration and tumor size. We have added the data (Figure 6F), and improved the manuscript.

(6) It is of great interest that the WNK-GID pathway causes β -catenin degradation independently of β TrCP. However, the physiological relationship between the GID complex and β TrCP in β -catenin degradation remains unclear. The authors should discuss this point.

As recommended by the reviewer, we have improved the manuscript by discussing about β TrCP and GID complex.

Thanks again for your constructive criticisms.

Sincerely,

Hiroshi Shibuya

REVIEWERS' COMMENTS:

Reviewer #1 (Remarks to the Author):

The paper has been revised adequately

Reviewer #2 (Remarks to the Author):

The authors have adequately addressed the initial concerns of this reviewer. The revised manuscript is suitable for publication in *Communication Biology* .